# Previous exposure to dengue virus is associated with increased Zika virus burden at the maternal-fetal interface in rhesus macaques

**Chelsea M. Crooks**[1], **Andrea M. Weiler**[2], **Sierra L. Rybarczyk**[2¤a], **Mason I. Bliss**[2], **Anna S. Jaeger**[3], **Megan E. Murphy**[4¤b], **Heather A. Simmons**[2], **Andres Mejia**[2], **Michael K. Fritsch**[5], **Jennifer M. Hayes**[2], **Jens C. Eickhoff**[6], **Ann M. Mitzey**[4], **Elaina Razo**[7], **Katarina M. Braun**[1], **Elizabeth A. Brown**[1], **Keisuke Yamamoto**[5¤c], **Phoenix M. Shepherd**[5], **Amber Possell**[2], **Kara Weaver**[2], **Kathleen M. Antony**[8], **Terry K. Morgan**[9,10], **Christina M. Newman**[5], **Dawn M. Dudley**[5], **Nancy Schultz-Darken**[2], **Eric Peterson**[2], **Leah C. Katzelnick**[11¤d], **Angel Balmaseda**[12], **Eva Harris**[11], **David H. O'Connor**[2,5], **Emma L. Mohr**[7], **Thaddeus G. Golos**[2,4,8], **Thomas C. Friedrich**[1,2]*, **Matthew T. Aliota**[3]*

**1** Department of Pathobiological Sciences, University of Wisconsin-Madison, Madison, Wisconsin, United States of America, **2** Wisconsin National Primate Research Center, University of Wisconsin-Madison, Madison, Wisconsin, United States of America, **3** Department of Veterinary and Biomedical Sciences, University of Minnesota, Twin Cities, St. Paul, Minnesota, United States of America, **4** Department of Comparative Biosciences, University of Wisconsin-Madison, Madison, Wisconsin, United States of America, **5** Department of Pathology and Laboratory Medicine, University of Wisconsin-Madison, Madison, Wisconsin, United States of America, **6** Department of Biostatistics and Medical Informatics, University of Wisconsin-Madison, Madison, Wisconsin, United States of America, **7** Department of Pediatrics, University of Wisconsin-Madison, Madison, Wisconsin, United States of America, **8** Department of Obstetrics and Gynecology, University of Wisconsin-Madison, Madison, Wisconsin, United States of America, **9** Department of Pathology, Oregon Health and Science University, Portland, Oregon, United States of America, **10** Department of Obstetrics and Gynecology, Oregon Health and Science University, Portland, Oregon, United States of America, **11** Division of Infectious Diseases and Vaccinology, University of California Berkeley, Berkeley, California, United States of America, **12** Sustainable Sciences Institute, Managua, Nicaragua

¤a Current address: Sangamo Therapeutics, Richmond, California, United States of America
¤b Current address: HIV Prevention Department, Terros Health, Phoenix, Arizona, United States of America
¤c Current address: Touro College of Osteopathic Medicine, New York, New York, United States of America
¤d Current address: National Institute of Allergy and Infectious Disease, Bethesda, Maryland, United States of America
* tfriedri@wisc.edu (TCF); mtaliota@umn.edu (MTA)

**Data Availability Statement:** All of the data used for figure generation and statistical analysis in this manuscript can also be found at https://github.

## Abstract

Concerns have arisen that pre-existing immunity to dengue virus (DENV) could enhance Zika virus (ZIKV) disease, due to the homology between ZIKV and DENV and the observation of antibody-dependent enhancement (ADE) among DENV serotypes. To date, no study has examined the impact of pre-existing DENV immunity on ZIKV pathogenesis during pregnancy in a translational non-human primate model. Here we show that macaques with a prior DENV-2 exposure had a higher burden of ZIKV vRNA in maternal-fetal interface tissues as compared to DENV-naive macaques. However, pre-existing DENV immunity had no detectable impact on ZIKV replication kinetics in maternal plasma, and all pregnancies progressed to term without adverse outcomes or gross fetal abnormalities detectable at

com/cmc0043/impact-of-denv-on-zikv-during-pregnancy-in-macaques. Raw FASTQ reads of the challenge stock of DENV-2/US/BID-V594/2006 are available at the Sequence Read Archive, BioProject accession number PRJNA435432. Raw FASTQ reads of the challenge stock of ZIKV PRVABC59 are available at the Sequence Read Archive, BioProject accession number PRJNA392686. The authors declare that all other data supporting the findings of this study are available within the article and its supplementary information files.

**Funding:** This work was supported by R01AI132563 from the National Institute of Allergy and Infectious Disease to M.T.A. and T.C.F and by P51OD011106 from the NIH Office of the Director. C.M.C. was supported by T32 AI007414 from the National Institute of Allergy and Infectious Disease. The funders had no role in study design, data collection and analysis, decision to publish, or preparation of the manuscript.

**Competing interests:** The authors have declared that no competing interests exist.

delivery. Understanding the risks of ADE to pregnant women worldwide is critical as vaccines against DENV and ZIKV are developed and licensed and as DENV and ZIKV continue to circulate.

## Author summary

Zika virus (ZIKV) gained global attention during an explosive outbreak in the Americas in 2015–16 when it was causally associated with the constellation of birth defects now termed congenital Zika syndrome (CZS). However, a substantial proportion of gestational ZIKV infections result in babies without apparent birth defects. Could there be other factors that influence ZIKV pathogenicity? For example, it is well-established that pre-existing immunity to one dengue virus (DENV) serotype can enhance the severity of a secondary DENV infection. ZIKV is antigenically closely related to DENV, but whether DENV-specific antibodies enhance the severity of ZIKV infection is unclear. To answer this question, we used our non-human primate model of ZIKV to assess the impact of pre-existing immunity to DENV on ZIKV pathogenesis during pregnancy. We did not observe any difference in ZIKV replication in plasma between macaques that were immune to DENV and those that were not. However, there was more ZIKV vRNA detected in the placenta of macaques immune to DENV, suggesting DENV immunity could enhance ZIKV infection of the placenta. As vaccines to both DENV and ZIKV are developed, it remains critical to understand the risks of DENV immunity for pregnant women exposed to ZIKV.

## Introduction

Pre-existing immunity to one DENV serotype can enhance the severity of a secondary heterologous DENV infection, a phenomenon known as antibody-dependent enhancement (ADE) [1–3]. ZIKV is genetically and antigenically closely related to DENV, raising the possibility that pre-existing DENV-specific antibodies might also modulate the severity of ZIKV infection. ADE is thought to occur when antibodies from a prior DENV infection bind to DENV virions and enhance uptake into Fcγ-receptor bearing cells rather than neutralizing viral infectivity. This can lead to increased viral replication, a more robust inflammatory response, and more severe disease [1,4,5].

Since the ZIKV outbreak in the Americas in 2015–2016, the potential role of DENV antibodies in ZIKV infection has been examined in a variety of *in-vitro*, *in-vivo*, and epidemiological studies. Studies in cell culture [6–16] and immunocompromised mice [6,7,13,17–19] have found a range of outcomes from enhancement of, to protection against, ZIKV infection. As measured by a DENV inhibition ELISA (iELISA) assay, an intermediate baseline DENV-specific antibody titer range of 1:21–1:80 was associated with a greater risk of developing severe dengue disease upon secondary exposure in a human cohort study [2]. In a separate human cohort study, a plaque reduction neutralization test (PRNT50) titer of <1:100 was associated with an increased risk of severe DENV disease upon secondary exposure [20].

Data from non-human primates (NHP) and human cohorts support the growing consensus that prior DENV infection does not enhance ZIKV infection in non-pregnant individuals [21–30]. However, DENV seroprevalence has been high in regions such as French Polynesia (>80%), Yap, and New Caledonia that subsequently experienced large-scale ZIKV outbreaks,

suggesting that high DENV seroprevalence does not protect against ZIKV outbreaks in a population [31–34].

The impact of prior DENV immunity on ZIKV pathogenesis during pregnancy remains unclear. Studies in placental macrophages [35], human placental explants [35–37], and both immunocompetent and immunocompromised pregnant mice [37,38] have all demonstrated enhancement of ZIKV infection in the presence of DENV antibodies. Retrospective studies of pregnant women in South America did not identify an association between DENV antibodies and adverse fetal outcomes [39–41]; however, a majority of women in these studies (>80%) had a prior DENV exposure, and outcomes could not be stratified by pre-existing anti-DENV titer. A retrospective study of microcephaly cases in Brazil indicated that there was reduced risk of microcephaly in areas with a DENV epidemic in the 6 years prior, but an increased risk of microcephaly in areas with a DENV epidemic >7 years prior, suggesting that the role of DENV-specific antibodies in modulating risk of congenital Zika syndrome (CZS) might change as antibody titers wane with time [42].

Understanding the potential impact of DENV immunity on ZIKV outcomes in pregnant women is critical, as vaccines against DENV and ZIKV are being developed, licensed, and distributed [43–45]. The rollout of Dengvaxia offers a cautionary tale, as vaccine-induced immunity led to more severe disease outcomes in seronegative individuals [46]. If ZIKV acts functionally as a fifth serotype of DENV, then one would expect that this vaccine would also enhance Zika disease by the same mechanism. Therefore, understanding whether the severity of maternal and fetal ZIKV infection increases in pregnant, DENV-immune individuals should be a public health priority.

NHP development and placentation resemble those of humans more closely than these processes do in other animal models, making NHPs particularly relevant to understanding viral infections in pregnancy [47]. Here we apply our established NHP model [48] to assess the impact of DENV immunity on ZIKV pathogenesis in pregnancy. We do not detect a role for DENV immunity in modulating fetal outcomes in ZIKV-infected pregnant macaques. However, previous exposure to DENV did appear to increase ZIKV vRNA burden in tissues of the maternal-fetal interface, a result that warrants further examination.

## Results

### Prior DENV immunity does not modulate ZIKV replication kinetics in plasma

To characterize the range of pathogenic outcomes of congenital ZIKV infection in DENV-immune animals, we subcutaneously (s.c.) inoculated a cohort of eight non-pregnant, Indian-origin rhesus macaques with $10^4$ PFU of DENV-2/US/BID-V594/2006, a low-passage human isolate from Puerto Rico (Fig 1). All eight macaques were productively infected with DENV-2, with peak plasma viral loads ranging from $10^5$–$10^7$ vRNA copies/mL occurring on days 2–3 post-inoculation (Fig 2). Following a biphasic decline in viral loads, all macaques cleared viremia by day 11 post-inoculation.

Macaques were bred 1–3 months following DENV inoculation. Once they became pregnant, the animals were challenged with $10^4$ PFU of ZIKV-PRVABC59 (ZIKV-PR), a human isolate from Puerto Rico, on gestational day 45. We chose this timepoint in the late first trimester since ZIKV infection during the first trimester is associated with the greatest risk of CZS in humans–it is a time of active neurological development for the fetus and is before many women know that they are pregnant [49,50]. ZIKV challenge was 84–119 days following DENV inoculation in each case. This cohort of eight DENV-immune macaques was compared to a cohort of four pregnant, DENV-naïve macaques that were inoculated with ZIKV-PR at

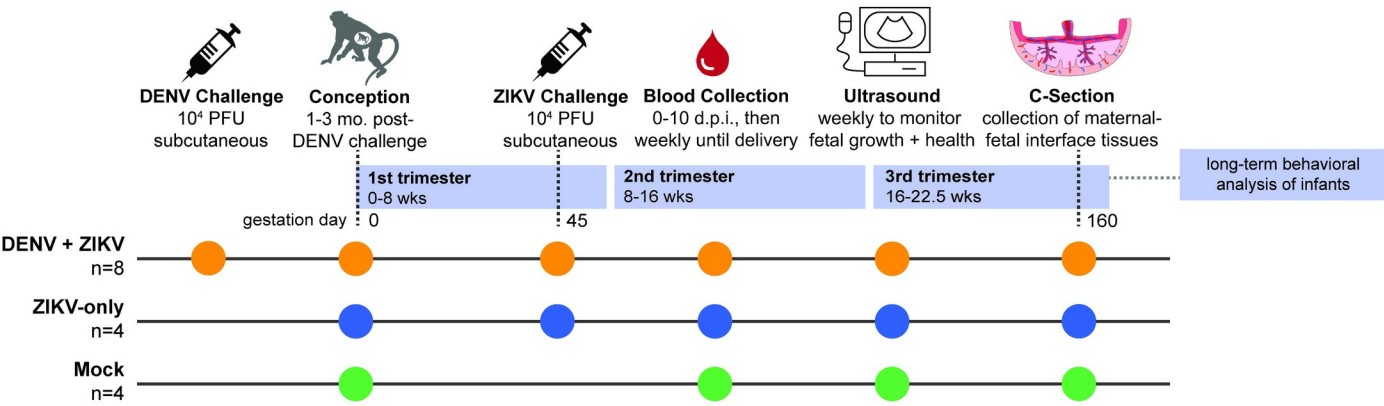

**Fig 1. Experimental Overview.** A cohort of eight non-pregnant macaques were challenged with $10^4$ PFU DENV-2 (orange). Approximately 1–3 months following DENV challenge, the eight DENV exposed macaques were bred, became pregnant, and were challenged with $10^4$ PFU ZIKV-PRVABC59, an Asian-lineage ZIKV isolate, on gestational day 45. A cohort of four pregnant, DENV-naïve macaques (blue) were challenged with ZIKV-PRVABC59 on gestational day 45. A control cohort of four macaques (green) were mock-challenged with PBS on gestational day 45. All three cohorts underwent the same experimental protocols for blood collection and sedation for ultrasound. At approximately gestational day 160, infants were delivered via cesarean section, and a set of maternal-fetal interface tissues with maternal biopsies were collected. Infants were placed with their mothers for long-term behavioral analysis, data from which is part of a separate study.

gestational day 45 and a cohort of four pregnant, DENV-naïve macaques mock-challenged with phosphate-buffered saline (PBS) at gestational day 45. Following challenge, all three cohorts (DENV-immune, DENV-naïve, and mock) underwent the same blood sampling and

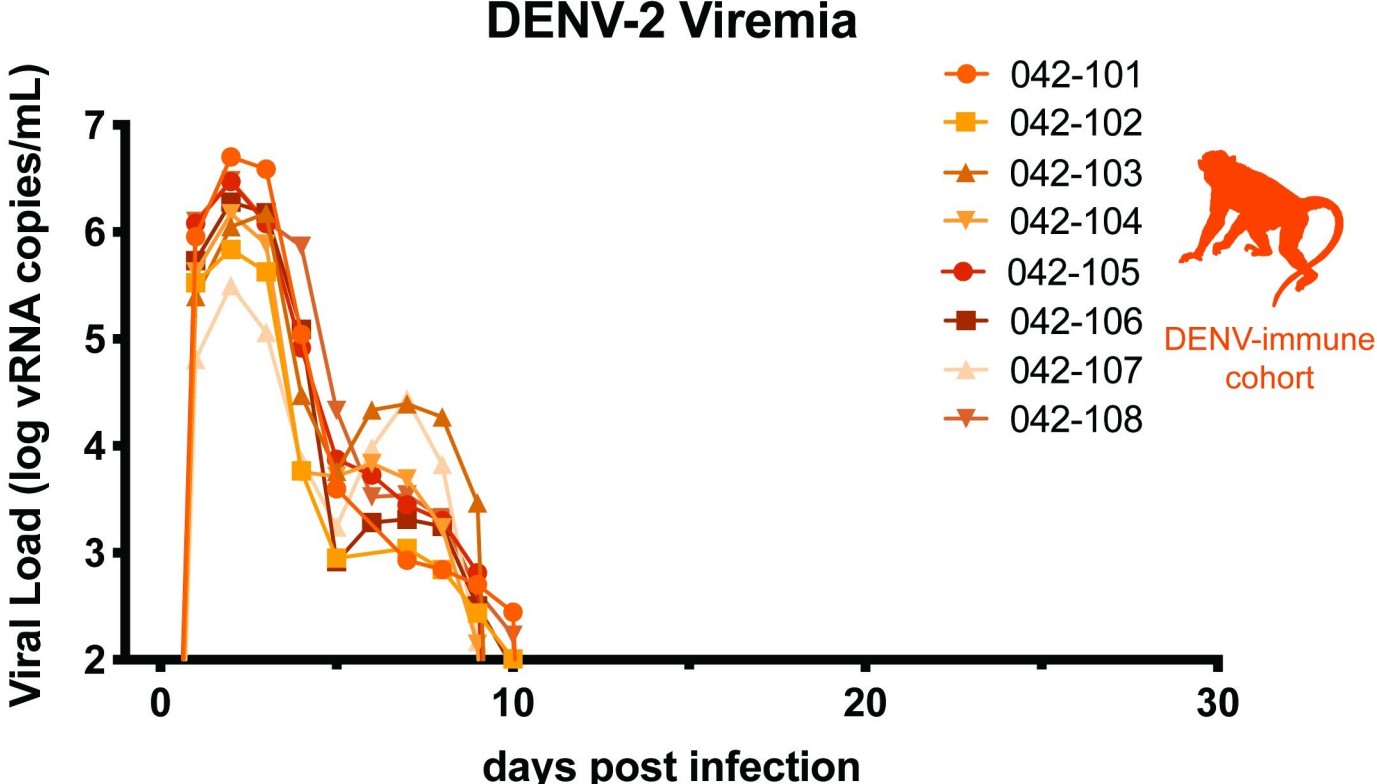

**Fig 2. Replication of DENV-2.** Eight non-pregnant macaques were inoculated with $10^4$ PFU DENV-2/US/BID-V594/2006, a 2006 human isolate from Puerto Rico. QRT-PCR was performed on plasma samples from 0–10, 14, 21, and 28 days post-infection (d.p.i). All values above the limit of quantification for the QRT-PCR assay (100 copies vRNA/mL plasma) are shown.

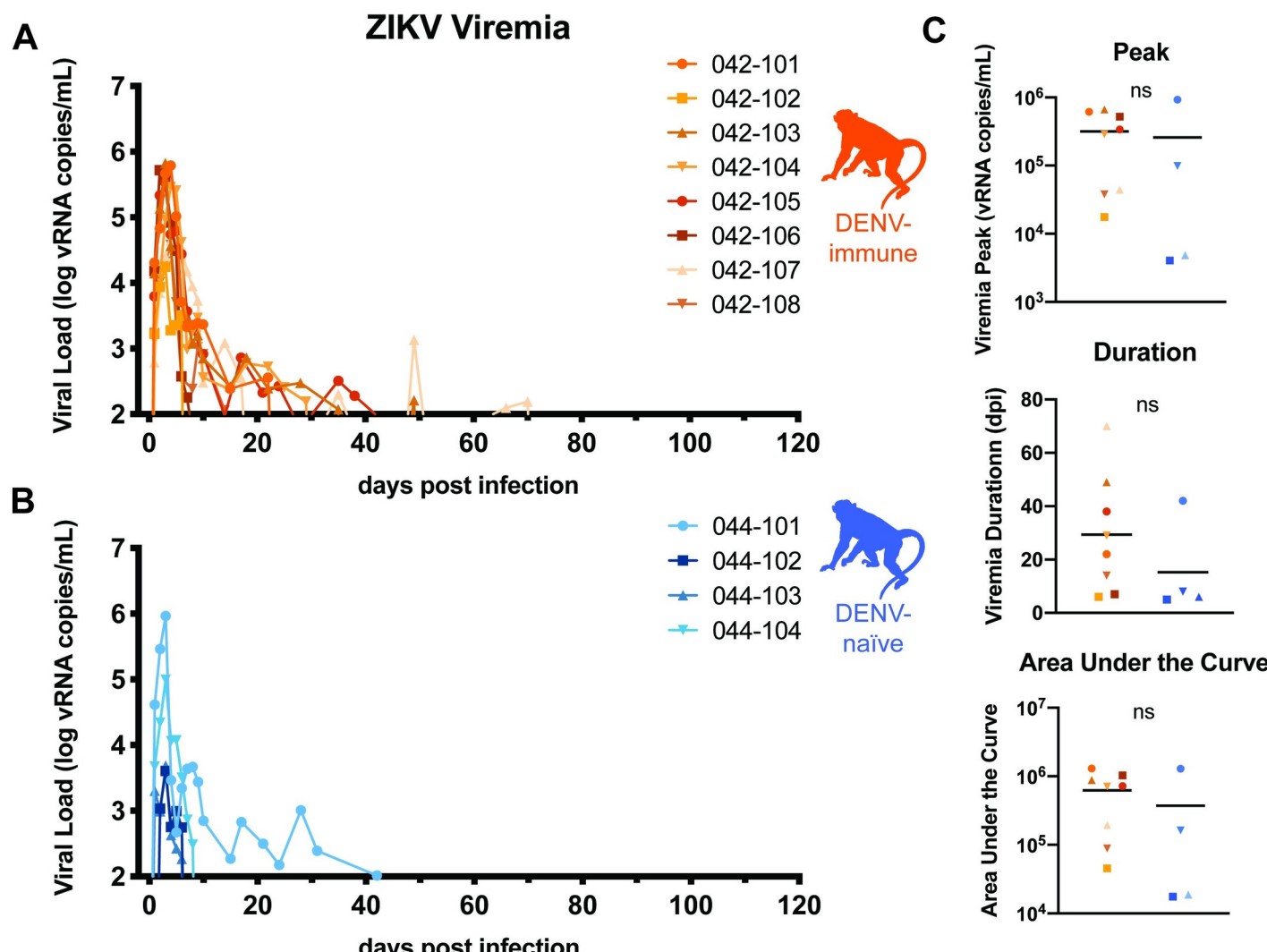

**Fig 3. ZIKV replication kinetics.** Eight DENV-immune (A, orange) and four DENV-naïve (B, blue) macaques were challenged with $10^4$ PFU of ZIKV-PRVABC59 at gestation day 45, which is late in the first trimester. Viral loads were assessed from plasma samples with ZIKV-specific QRT-PCR. All values above the limit of quantification (100 copies vRNA/mL plasma) are shown above. C. Graphs of the values for the peak, duration, and area under the curve of viremia for both DENV-immune and DENV-naïve macaques. An unpaired t-test was used for statistical comparison; ns = not significant (p > 0.05). Only values above the limit of quantification were used in statistical analyses.

fetal monitoring protocols (Fig 1). All macaques inoculated with ZIKV were productively infected. Peak plasma viremia occurred on days 2–4 post-challenge, with titers ranging from $10^4$–$10^5$ vRNA copies/mL in DENV-immune animals and $10^3$–$10^5$ vRNA copies/mL in DENV-naïve animals (Fig 3A and 3B). An unpaired t-test did not reveal significant differences between cohorts in the peak, area under the curve, or duration of viremia (Fig 3C). Since prolonged ZIKV viremia >21 days is only observed in pregnancy, we assessed differences in duration both as a continuous variable and as a binary with viremia greater than or less than 21 days. This suggests that prior DENV-2 immunity did not alter ZIKV replication kinetics during gestation.

## DENV-immune macaques have low levels of ZIKV cross-reactive antibodies present at the time of challenge

In order to assess how cross-reactive DENV antibodies impact ZIKV outcomes during pregnancy, we characterized DENV and ZIKV antibody dynamics throughout the experimental time course. We collected serum samples from macaques at 28 days post-DENV challenge, the day of ZIKV challenge, 28–35 days post-ZIKV challenge, and the day of c-section for measuring antibody responses to DENV and ZIKV. We used PRNT and iELISA to measure neutralizing antibodies or binding antibodies, respectively. In the PRNT, serial dilutions of serum antibodies are incubated with DENV or ZIKV, plated on a confluent monolayer of cells, and assessed for the dilution of antibodies required to reduce plaques by 50 or 90 percent (S1 Fig). In iELISA, serum is serially diluted with peroxidase-conjugated DENV- or ZIKV-specific antibodies, which compete for binding to either an equal mixture of DENV1-4 antigens or ZIKV antigen [2,51]. Due to the impact of COVID-19, only 4 of 8 DENV-immune macaques were assayed via iELISA.

At 28 days post-DENV challenge, all eight macaques seroconverted and developed a robust antibody response to DENV-2 as measured by both DENV PRNT and DENV iELISA (Fig 4A, 4C, and 4D). At this time point, all macaques also showed a cross-reactive antibody response to ZIKV in one or both assays (3 of 4 macaques by iELISA and 7 of 8 macaques by PRNT), although generally below levels considered to be protective against subsequent ZIKV challenge (Fig 4B, 4E, and 4F)[52].

At the time of the ZIKV challenge, 84–119 days after DENV inoculation, DENV antibody titers had increased four-fold in 6 of the 8 DENV-exposed macaques by PRNT and 4 of 4 macaques by iELISA as compared to titers at 28 days post-DENV infection (Fig 4G, 4I, and 4J). Cross-reactive ZIKV antibody titers generally remained stable or increased only modestly (Fig 4H, 4K, and 4L) as compared to titers at 28 days post-DENV infection in the majority of macaques. However, cross-reactive ZIKV antibodies became undetectable by PRNT in 3 of 4 macaques that previously showed cross-reactivity at 28 days post-DENV challenge (Fig 4K and 4L). By using both assays, we detected low levels of cross-reactive antibodies to ZIKV at the time of ZIKV challenge in all DENV-immune macaques. In fact, 2 of 4 macaques had ZIKV iELISA titers that fell within the range 1:21–1:80, which has previously been shown to increase risk of more severe DENV disease in humans [2]. The DENV and ZIKV antibody titers on the day of ZIKV challenge were not correlated with the time between DENV and ZIKV infections (r2 < 0.1). At the time of ZIKV challenge, no antibody responses to either ZIKV or DENV were detected using either assay in the DENV-naïve macaques (Fig 4G, 4H, 4K, and 4L).

Between 28–35 days post-ZIKV challenge, DENV antibody titers increased approximately four-fold following ZIKV challenge in DENV-immune macaques, as assessed by both DENV iELISA and PRNT (Fig 4M, 4O, and 4P). DENV titers in DENV-naïve macaques were only assessed via DENV iELISA, which revealed essentially no evidence of cross-reactive DENV antibodies, with a low-level antibody titer (1:11) to DENV in only 1 of 4 macaques (Fig 4M). By 28–35 days post-ZIKV challenge, both DENV-immune and DENV-naïve macaques developed robust ZIKV-specific responses as measured by both ZIKV iELISA and PRNT (Fig 4N, 4Q, and 4R). Macaques in both cohorts that had viremia for a duration of >21 days (042–101, 042–103, 042–104, 044–101) developed antibody titers more than four-fold higher than those animals that had viremia for a duration of <21 days (042–102, 044–102, 044–103, 044–104) as determined by ZIKV iELISA. PRNT50 titers were significantly higher (p = 0.0095) in DENV-immune macaques than DENV-naïve animals 28–35 days after ZIKV challenge, but no significant differences were noted in PRNT90 titers between groups. Together, these data provide evidence that antibodies capable of cross-reacting with ZIKV were present at the time of ZIKV

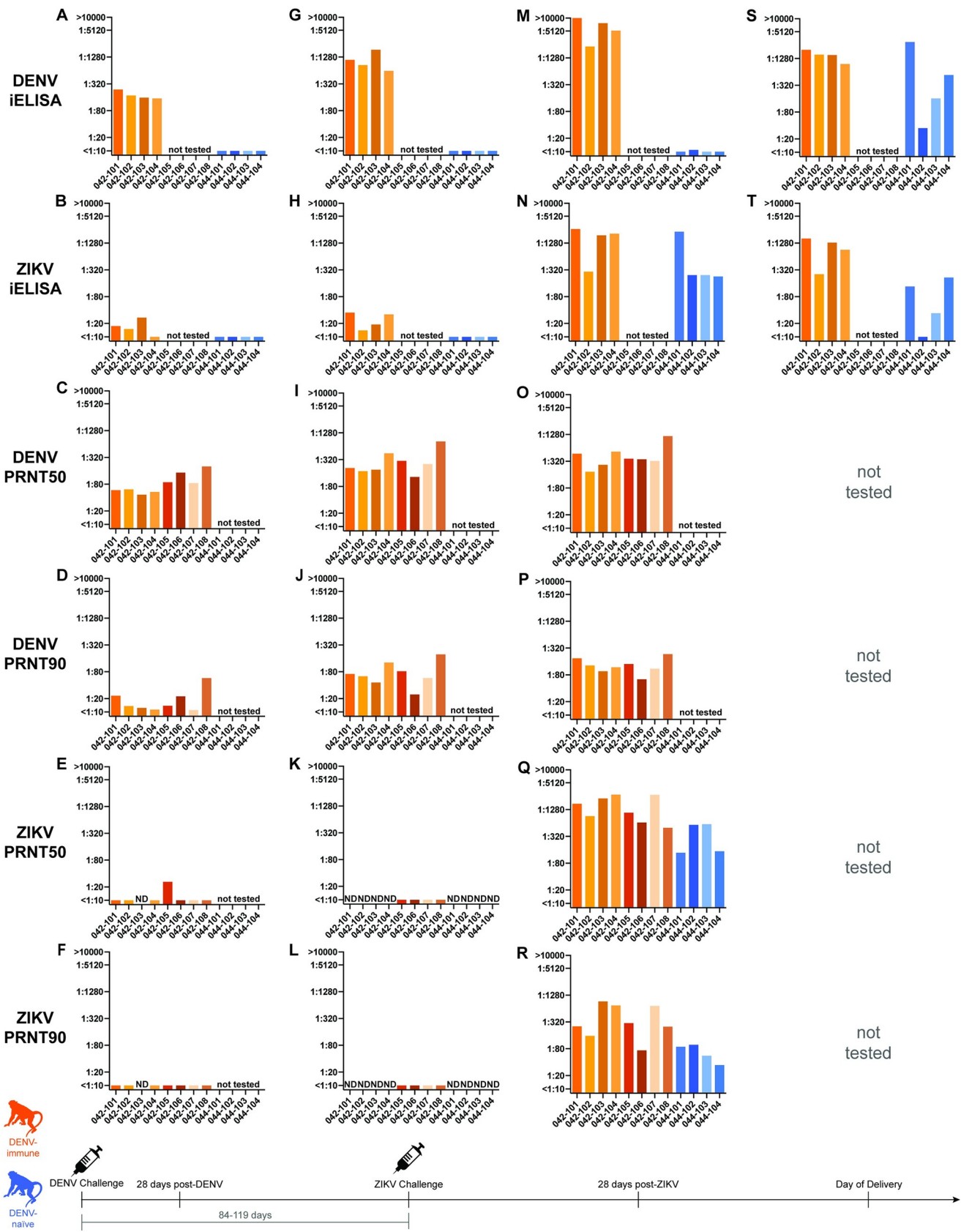

**Fig 4. DENV and ZIKV antibody dynamics.** iELISA and PRNT titers against DENV and ZIKV 28 days post-DENV inoculation (A-F), the day of ZIKV challenge (G-L), 28–35 days post-ZIKV challenge (M-Q), and the day of delivery (S-T). iELISA and PRNT titers from DENV-immune animals shown in orange and from DENV-naïve animals shown in blue. Samples labeled "ND" were not detected. Using an unpaired t-test, PRNT50, but not PRNT90, titers from the DENV-immune group were significantly higher than the PRNT50 titer of DENV-naïve animals at 28 days post-ZIKV-challenge (**p<0.01). Neutralization curves can be found in S1 Fig.

challenge in DENV immune animals and show that all animals, regardless of DENV exposure history, develop a robust antibody response to ZIKV.

## No evidence of fetal growth restriction during gestation

To further characterize pathogenic outcomes during pregnancy, we define fetal health and growth parameters throughout gestation. No gross abnormalities, such as microcephaly, missing limbs, or hydrops fetalis were noted in any animals during gestation. Head circumference and biparietal diameter measurements were used to assess head size; femur length and abdominal circumference were used to assess overall fetal growth. Fetal measurements were compared to previously collected normative data on fetal growth trajectories in 55 pregnant rhesus macaques [53,54]. A z-score (number of standard deviations from the normative data) was calculated for each measurement at each timepoint. To account for animal-specific differences, z-scores were plotted as the change from the baseline measurement (open circles, Fig 5). Overall group growth trajectories were calculated (solid line, Fig 5) and used for statistical comparisons. Only the biparietal diameter of the mock-infected cohort was significantly lower than the normative data (p = 0.01713). There were no significant differences noted in pairwise comparisons of growth trajectories between groups. Taken together, these extensive fetal growth measurements suggest that there was no significant reduction in fetal growth in ZIKV-exposed macaques, regardless of their DENV immune history.

## No evidence of vertical transmission in either DENV-immune or DENV-naïve macaques

At approximately gestational day 160 (term = gestational day 165), infants were delivered via cesarean section. During the surgery, a biopsy of the maternal mesenteric lymph node was taken to look for ZIKV vRNA in the dam. None of the mesenteric lymph node biopsies were positive in the DENV-immune cohort and only one of four mesenteric lymph node biopsies was positive in the DENV-naïve cohort, a difference which was not significant (S1 Table). Fetal tissues are not available for virological analysis because infants were placed with their mothers for long-term behavioral analysis; results from this ongoing study, which includes longitudinal serology, will be the subject of a future report. We collected fetal plasma, umbilical cord plasma, and amniotic fluid; none of the fluid samples from infants in either cohort tested positive for ZIKV vRNA (S1 Table). Additionally, we could not detect ZIKV-specific IgM in serum from any neonate on the day of delivery (S2 Fig). Synthesizing our data, we found no evidence of direct infection of the fetus in either cohort, although the possibility of vertical transmission with viral clearance by the time of delivery cannot be ruled out.

## Higher vRNA burden in the maternal-fetal interface in DENV-immune macaques

We performed an extensive dissection of both discs of the placenta in order to understand the distribution of ZIKV in placental tissues. Positive tissue samples were detected above the theoretical limit of detection of our QRT-PCR assay in 5 of 8 DENV-immune macaques and only 1 of 4 DENV-naïve macaques (Fig 6A). Using a Mann-Whitney U test, there was a significantly

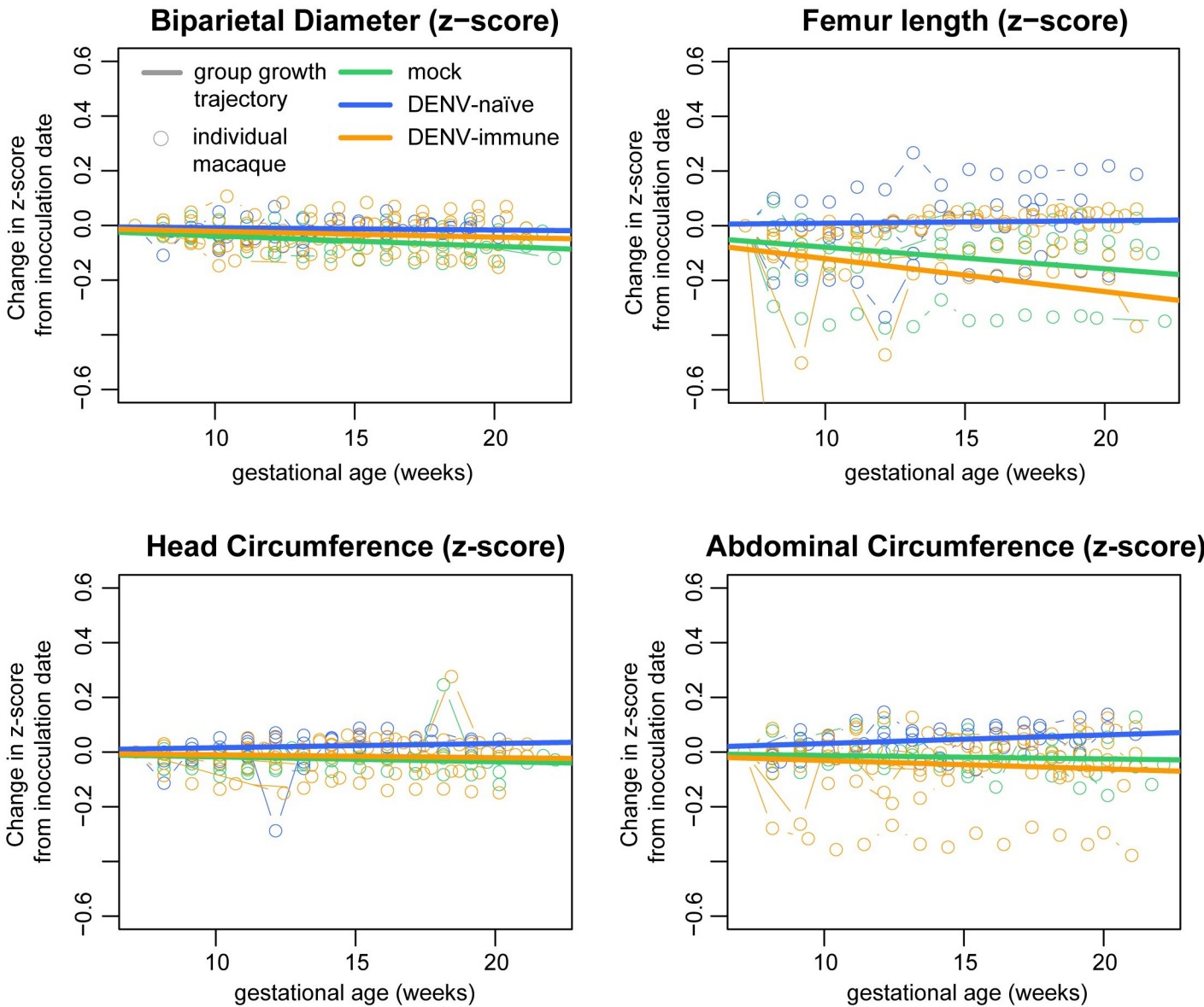

**Fig 5. Fetal Growth.** Comprehensive ultrasounds were performed weekly throughout gestation to monitor fetal health and perform four measurements of fetal growth: biparietal diameter and head circumference to evaluate head size; abdominal circumference and femur length to evaluate overall fetal growth. Using normative data from the California National Primate Research Center, a z-score was calculated for each measurement and the change in z-score from baseline is plotted for each measurement with an open circle. The overall growth trajectory for each group was quantified by calculating the regression slope parameters from baseline (solid line). When compared to the normative data, mock-infected animals had significantly reduced biparietal diameter growth (p = 0.01713). No other significant differences were detected in comparisons to the normative data or in comparisons between the experimental groups.

higher burden of ZIKV RNA in the chorionic villi in the DENV-immune group as compared to the DENV-naïve group (p<0.01). Although there was a trend toward a greater burden of ZIKV in the fetal membranes in DENV-immune macaques, there were no statistically significant differences between cohorts in vRNA burden in the other MFI tissues (decidua, chorionic plate, umbilical cord, fetal membranes, and uterine placental bed). The highest ZIKV RNA burden detected in a fetal membrane sample was from DENV-immune animal 042–104, which had a viral load of $1.03 \times 10^5$ vRNA copies/ml. We could not recover infectious virus

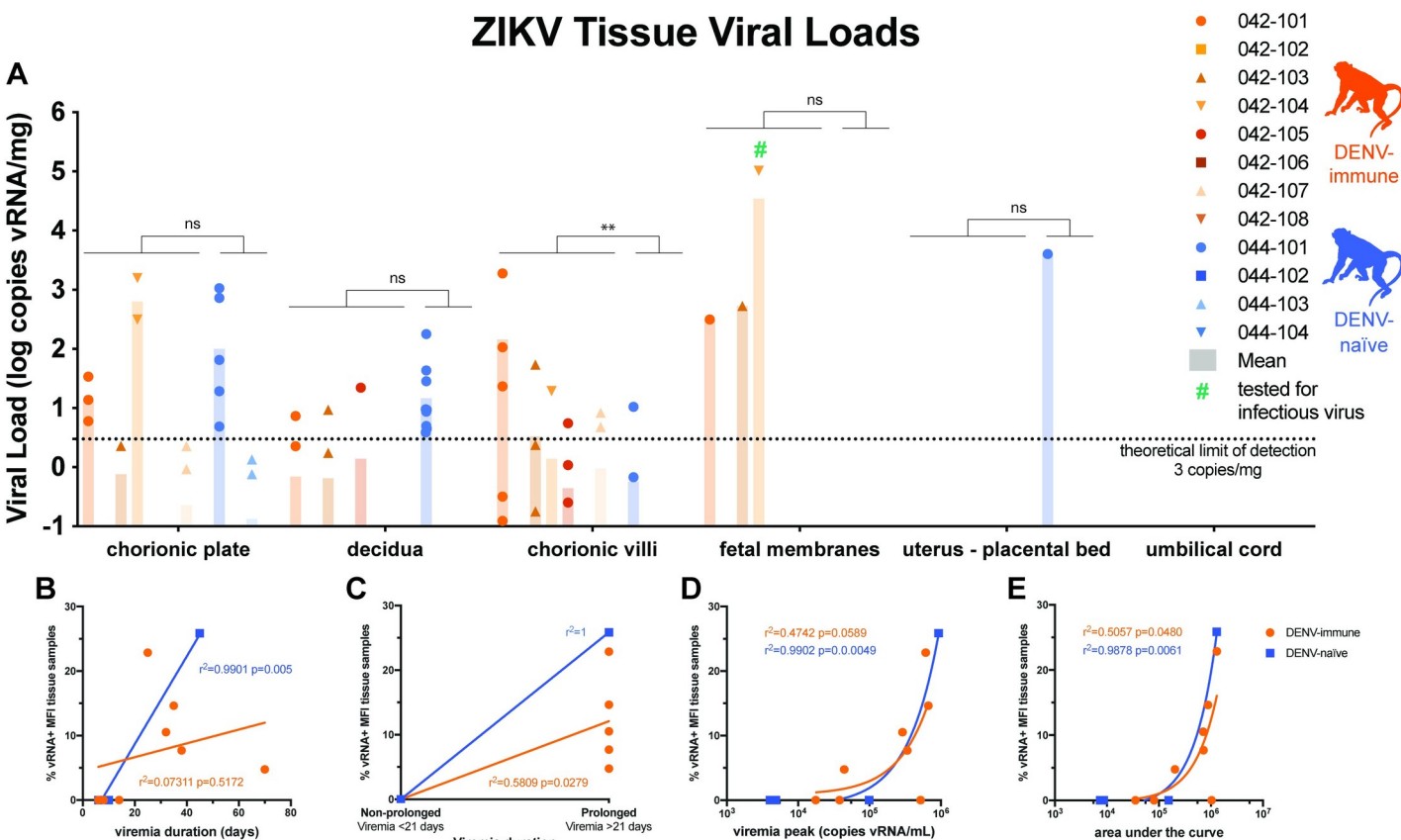

**Fig 6. Maternal-Fetal Interface Viral Loads.** All tissue samples were tested for the presence of viral RNA using ZIKV-specific QRT-PCR. **A.** All tissues >0.1 copy vRNA/mg tissue are shown above; only tissues with viral loads greater than the theoretical limit of quantification (3 copies vRNA/mg) were used for statistical analysis. A Mann-Whitney U test was used to assess statistically significant differences between the experimental groups (**p<0.01). **B-E.** Pearson correlation analysis was performed to assess correlation between the percent of tissues collected that were vRNA positive and the duration (**B** and **C**), peak (**D**), and area under the curve (**E**) of viremia.

from this specimen; we did not attempt virus isolation from other specimens, which had much lower viral loads ($<10^3$ copies/mg).

To determine whether the presence of vRNA in the MFI was associated with duration, peak, or area under the curve of viremia, we performed a Pearson correlation analysis. When prolonged viremia was defined as >21 days and non-prolonged viremia as <21 days (Fig 6B) there was a significant positive correlation between prolonged viremia and presence of vRNA in the MFI for both the DENV-immune and DENV-naïve cohorts. When viremia is assessed as a continuous variable, the correlation is no longer significant for the DENV-immune cohort (Fig 6C). There was a significant correlation between area under the curve and presence of vRNA in the MFI in both cohorts (Fig 6D and 6E). There was a significant correlation between peak viremia and presence of vRNA in the MFI only in DENV-naïve animals (Fig 6E).

## More-severe histopathological changes inconsistently detected in DENV-immune macaques

Placental insufficiency due to virus-mediated damage could lead to poor fetal outcomes [55]. In order to assess the impact of ZIKV infection on MFI health, we quantified inflammation and infarctions within the MFI. Qualitative pathological findings included transmural

**Table 1. Placental cotyledon pathology.**

| Group | Dam | % CHIV + cotyledons | Infarcted cotyledons/total cotyledons (%) | Villous stromal calcifications (present/absent) | Vasculopathy (present/absent) | Placental weight (g) |
|---|---|---|---|---|---|---|
| Mock | 044–105 | 0.0 | 5.88 | Present | Absent | 111.08 |
| | 044–106 | 0.0 | 12.5 | Present | Absent | 106.5 |
| | 044–107 | 0.0 | 0.0 | Present | Present | 144.48 |
| | 044–108 | 0.0 | 45.5 | Present | Absent | 122.92 |
| DENV-naïve | 044–101 | 0.0 | 25.0 | Present | Absent | 172.59 |
| | 044–102 | 0.0 | 33.3 | Present | Absent | 123.87 |
| | 044–103 | 0.0 | 0.0 | Absent | Absent | 134.49 |
| | 044–104 | 0.0 | 18.2 | Absent | Absent | 120.48 |
| DENV-immune | 042–101 | 0.0 | 21.43 | Present | Absent | 104.4 |
| | 042–102 | 7.69 | 7.69 | Present | Absent | 111.9 |
| | 042–103 | 0.0 | 0.00 | Present | Absent | 120.06 |
| | 042–104 | 0.0 | 26.67 | Present | Absent | 95.33 |
| | 042–105 | 0.0 | 25.00 | Absent | Absent | 119.97 |
| | 042–106 | 0.0 | 53.33 | Present | Present | 120.14 |
| | 042–107 | 0.0 | 28.57 | Present | Absent | 139.74 |
| | 042–108 | 0.0 | 33.33 | Present | Absent | 129.54 |

infarctions and neutrophilic deciduitis in the central cross-section of both placental discs examined, but these findings were observed in animals of all groups, including mock-infected animals, with no consistent patterns distinguishing groups. In order to quantitatively analyze placental pathology and identify any trends within and between cohorts, the center section of each placental disc was scored for 22 pathologic changes associated with fetal vascular malperfusion, maternal vascular malperfusion, and generalized placental disease (S2 Table). DENV-immune macaques had significantly higher scores in four pathologic changes in disc 1 (% transmural infarction, chronic villitis, avascular villi, and chronic retroplacental hemorrhage) and one pathologic change in disc 2 (chronic villitis) as compared to the mock-infected cohort (S3 Fig) There were no significant differences between DENV-naïve animals and mock-infected animals.

We also assessed a cross-section of each of the individual placental cotyledons, including the decidua basalis, for the presence of chronic histiocytic intervillositis (CHIV), infarctions, villous stromal calcifications, and vasculopathy (Table 1). Although infarctions and villous stromal calcifications were present in DENV-immune and DENV-naïve macaques, they were also present in mock-infected animals. There were no statistically significant differences between any of the groups for any of these pathologic features or placental weight. This suggests that the presence of some changes, such as multifocal areas of villous mineralization, may

be a result of normal placental aging or a result of stress from experimental procedures, rather than from viral infection. These data underscore the necessity of mock-infected controls when assessing pathology.

## Discussion

This study provides the first comprehensive assessment of the impact of pre-existing DENV immunity on ZIKV pathogenesis during pregnancy in a translational NHP model. Macaques with previous DENV-2 exposure supported robust replication of ZIKV and developed a robust neutralizing antibody response to ZIKV, suggesting that primary DENV-2 infection had no protective effect. We did not observe evidence of enhanced ZIKV replication in DENV-immune macaques as compared to DENV-naïve macaques. Neither intrauterine growth restriction nor adverse fetal outcomes were observed in either cohort. However, we did observe ZIKV RNA in the MFI in a greater number of DENV-immune macaques and a significantly greater burden of ZIKV RNA in the chorionic villi in DENV-immune macaques as compared to DENV-naïve macaques. Although we do not have any evidence of direct fetal infection, ZIKV vRNA was detected in the chorionic plate, the fetal side of the placenta, in three macaques [56]. Enhanced infection of the chorionic villi is consistent with prior studies that have shown increased replication of ZIKV in the placenta of mice and placental cells in the presence of DENV antibodies [35,37,38]. The implications of increased infection of the placenta on fetal outcomes is unclear, since we observed no fetal demise nor any of the other clinical sequelae associated with CZS in offspring. This also suggests that the presence of ZIKV in the maternal-fetal interface may not be a robust indicator of significant fetal harm in this model. Future studies will define the effects of DENV and ZIKV on longer-term infant neurobehavioral development, as developmental deficits are the most common adverse outcome of prenatal ZIKV exposure in humans [57].

We did observe an association between prolonged viremia, defined as lasting >21 days, and the presence of ZIKV vRNA in the maternal-fetal interface. Since 5 of 8 DENV-immune macaques had viremia greater than 21 days, while only 1 of 4 DENV-naïve animals did, it is tempting to speculate that prior DENV immunity may lead to longer viral replication and greater ZIKV burden in the placenta. Specifically, in the chorionic villi where we detected a significant difference in vRNA burden between groups, animals with positive tissue samples all had viremia >21 days. However, since we did not observe any statistically significant differences in the duration of viremia between the two groups, perhaps due to a small sample size, we cannot make any definitive conclusions about the impact of prior DENV immunity on the duration of ZIKV viremia. In thinking about why we observed an increase in vRNA burden the placenta in the absence of increased viremia, it is conceivable that pre-existing immunity could enhance viral burdens in tissues without necessarily detectably increasing viremia, for example if ADE specifically enhanced infection of FcR-bearing cells in the placenta. Because we were able to collect multiple placental samples from each macaque, this also increased our power to detect differences in tissue burden in the absence of a difference in viremia. We did correlate prolonged viremia to presence of virus in the MFI, which may be a marker for a process that facilitates enhanced infection of the MFI. Or perhaps clearance from MFI tissues is delayed relative to clearance of virus from plasma, such that we can detect higher levels of MFI tissue burden remaining at birth in dams who had prolonged viremia. The lack of enhanced viremia does not preclude the possibility of more severe fetal outcomes.

A significant strength of this study was our ability to assess ZIKV pathogenesis in a translational model in macaques with known infection histories. This allowed us to report detailed antibody dynamics throughout the course of infection, historical data that can be challenging to obtain in human cohort studies particularly during pregnancy. We confirmed the presence

of low levels of cross-reactive antibodies present at the time of ZIKV challenge in our DENV-immune cohort. Twenty-eight days after ZIKV-challenge, we determined that PRNT50, but not PRNT90, titers were significantly higher in our DENV-immune cohort. We were particularly interested in this finding, since a higher ZIKV neutralization titer at the time of delivery has been associated with CZS in human cohort studies [40]. However, at the time of delivery there were no significant differences in iELISA titers between cohorts.

As is common to non-human primate studies, ethical and financial constraints limited the number of variables that we were able to test in this study. A significant limitation of this study is the small group sizes used. Since the most severe effects of ZIKV only occur in a minority of cases, it is difficult to model the full spectrum of disease that women experience when infected with ZIKV during pregnancy. Small group sizes further limited our statistical power to detect significant differences between groups. The group sizes used here resulted in a 60% power level to detect a moderate effect size of d = 1.5. In this study, we only tested a single DENV serotype; there is considerable evidence that the sequence of infecting DENV serotypes has an effect on subsequent enhancement or protection (for review see [21]). We chose this serotype since it has been associated with enhancement among serotypes and is highly prevalent in the region where the most recent ZIKV epidemic occurred [21,58]. As is common to non-human primate studies, we did not expect to see any clinical signs of infection such as rash, arthralgia or fever at this physiologically relevant dose; however, this DENV-2 isolate, which has not previously been used in a non-human primate study, did exhibit robust replication and a strong antibody response. There is also considerable evidence that the pre-existing antibody titer at the time of secondary infection is associated with the risk of developing severe disease [2,20]. In this study, we had a relatively short window (1–3 months) between DENV and ZIKV inoculations, and a different interval might have affected the titer of cross-reactive antibodies present at the time of ZIKV challenge.

We tested a single ZIKV isolate, dose, and inoculation time point in gestation; changes to any of these parameters could have elicited more significant differences in maternal or fetal outcomes. We chose to use ZIKV-PR since it is a widely used contemporary isolate from the Americas. In studies performed using this isolate at the Wisconsin National Primate Research Center, our group has observed outcomes ranging from fetal demise to, more commonly, no detectable impact. A cross-center study estimated that, in the specific model used here, adverse fetal outcomes occur in approximately 1 in 7 pregnancies [59]. After our study began, another group reported an amino acid change common to laboratory-passaged ZIKV-PR stocks that might attenuate viral pathogenicity in mice [60]. We cannot determine whether using a ZIKV stock with a different sequence would have affected the results we report here.

The relationship between flavivirus antibodies and disease outcomes is complex, depending on factors including antibody titer, specificity, and degree of sequence conservation among viruses. It is therefore difficult to comprehensively disentangle all these factors in a single experiment. More work is needed to understand the relationship between DENV immunity, viral infection of the placenta, and prolonged viremia. While there is a growing consensus that DENV may not enhance ZIKV in non-pregnant individuals, this study provides evidence that more research is needed to understand the risks associated with prior DENV immunity on ZIKV pathogenesis in pregnancy.

## Methods

### Ethics statement

This study was approved by the University of Wisconsin College of Letters and Sciences and Vice Chancellor for Research and Graduate Education Centers Institutional Animal Care and Use Committee (Approved protocol numbers: G005401 and G006139).

## Experimental design

This study was designed to assess the impact of pre-existing DENV immunity on ZIKV pathogenesis during pregnancy in a non-human primate model. Eight female non-pregnant Indian origin rhesus macaques (*Macaca mulatta*) were inoculated subcutaneously with $1x10^4$ PFU of DENV-2/US/BID-V594/2006. Approximately 1–3 months following DENV challenge, macaques were bred and became pregnant. All eight macaques were then inoculated subcutaneously with $1x10^4$ PFU of ZIKV-PRVABC59 (ZIKV-PR) between 44–50 days of gestation (term is 165 ± 10 days). Macaques were monitored throughout the remainder of gestation. At approximately gestation day 160, infants were delivered via cesarean section and monitored for long-term development. A comprehensive set of maternal biopsies and maternal-fetal interface were collected for analysis. For the DENV-naïve group, four pregnant Indian origin rhesus macaques (*Macaca mulatta*) were inoculated subcutaneously with $1x10^4$ PFU of ZIKV-PR between 44–50 days of gestation (term is 165 ± 10 days). Macaques were monitored throughout the remainder of gestation. At approximately gestation day 160, infants were delivered via cesarean section and monitored for long-term development. A comprehensive set of maternal biopsies and maternal-fetal interface were collected for analysis. A cohort of four pregnant PBS-inoculated animals served as a control group and underwent the same experimental regimen, including the sedation for all blood draws and ultrasounds, as the ZIKV-infected cohort. In order to minimize the number of animals used in studies of ZIKV pathogenesis, the DENV-naïve and mock-infected cohort have served as a control group for other studies [61].

## Care and use of macaques

All macaque monkeys used in this study were cared for by the staff at the WNPRC in accordance with the regulations and guidelines outlined in the Animal Welfare Act and the Guide for the Care and Use of Laboratory Animals and the recommendations of the Weatherall report (https://royalsociety.org/topics-policy/publications/2006/weatherall-report/). All macaques used in the study were free of *Macacine herpesvirus 1*, simian retrovirus type D (SRV), simian T-lymphotropic virus type 1 (STLV), and simian immunodeficiency virus (SIV). For all procedures (including physical examinations, virus inoculations, ultrasound examinations, and blood collection), animals were anaesthetized with an intramuscular dose of ketamine (10 mg/kg). Blood samples were obtained using a vacutainer system or needle and syringe from the femoral or saphenous vein.

## Cells and viruses

DENV-2/US/BID-V594/2006 was originally isolated from a human in Puerto Rico with one round of amplification on C6/36 cells. This DENV-2 isolate was obtained from BEI resources (NR-43280, Manassas, VA). Zika-virus/H.sapiens-tc/PUR/2015/PRVABC59_v3c2 (ZIKV-PR) was originally isolated from a human in Puerto Rico in 2015, with three rounds of amplification on Vero cells, was obtained from Brandy Russell (CDC, Fort Collins, CO, USA). African Green Monkey kidney cells (Vero; ATCC #CCL-81) were maintained in Dulbecco's modified Eagle medium (DMEM) supplemented with 10% fetal bovine serum (FBS; Hyclone, Logan, UT), 2 mM L-glutamine, 1.5 g/L sodium bicarbonate, 100 U/ml penicillin, 100 μg/ml of streptomycin, and incubated at 37°C in 5% $CO_2$. *Aedes albopictus* mosquito cells (C6/36; ATCC #CRL-1660) were maintained in DMEM supplemented with 10% fetal bovine serum (FBS; Hyclone, Logan, UT), 2mM L-glutamine, 1.5 g/L sodium bicarbonate, 100 U/ml penicillin, 100 μg/ml of streptomycin, and incubated at 28°C in 5% $CO_2$. The cell lines were obtained from the American Type Culture Collection, were not further authenticated, and were not

specifically tested for mycoplasma. Virus stocks were prepared by inoculation onto a confluent monolayer of C6/36 cells; a single, clarified stock was harvested for each virus, with a titer of 1.55 x $10^5$ PFU/ml for DENV-2 and 1.58 x $10^7$ PFU/ml for ZIKV-PR. Deep sequencing with limited PCR cycles confirmed that the DENV-2 virus stock was identical to the reported sequence in GenBank (EU482725) at the consensus level. Twelve nucleotide variants were detected at 5.3–16.1% frequency. Amplicon deep sequencing of ZIKV-PR virus stock using the methods described in Quick, et al. [62] revealed two consensus-level nucleotide substitutions in the stock as compared to the reported sequence in GenBank (KU501215), as well as seven other minor nucleotide variants detected at 5.3–30.6% frequency. Details on accessing sequence data can be found in the Data Accessibility section.

## Plaque assay

All titrations for virus quantification from virus stocks and screens for infectious ZIKV from macaque tissue were completed by plaque assay on Vero cell cultures as previously described [63]. Briefly, duplicate wells were infected with 0.1 ml aliquots from serial 10-fold dilutions in growth media and virus was adsorbed for one hour. Following incubation, the inoculum was removed, and monolayers were overlaid with 3ml containing a 1:1 mixture of 1.2% oxoid agar and 2X DMEM (Gibco, Carlsbad, CA) with 10% (vol/vol) FBS and 2% (vol/vol) penicillin/ streptomycin (100 U/ml penicillin, 100 µg/ml of streptomycin). Cells were incubated at 37˚C in 5% $CO_2$ for four days for plaque development. Cell monolayers were then stained with 3 ml of overlay containing a 1:1 mixture of 1.2% oxoid agar and 2X DMEM with 2% (vol/vol) FBS, 2% (vol/vol) penicillin/streptomycin, and 0.33% neutral red (Gibco). Cells were incubated overnight at 37˚C and plaques were counted.

## Inoculations

Inocula were prepared from a viral stock propagated on a confluent monolayer of C6/36 cells. The stocks were thawed, diluted in PBS to $10^4$ PFU/ml and loaded into a 1 mL syringe that was kept on ice until challenge. Animals were anesthetized as described above and 1 ml of inocula was delivered subcutaneously over the cranial dorsum. Animals were monitored closely following inoculation for any signs of an adverse reaction.

## Ultrasound measurements

Ultrasound measurements were taken according to the procedures described previously [48]. Briefly, dams were sedated with ketamine hydrochloride (10mg/kg) for weekly sonographic assessment to monitor the health of the fetus (heart rate) and to take fetal growth measurements, including the fetal femur length (FL), biparietal diameter (BPD), head circumference (HC), and abdominal circumference (AC). Weekly fetal measurements were plotted against mean measurement values and standard deviations for fetal macaques developed at the California National Primate Research Center [53,54]. Additional Doppler ultrasounds were taken as requested by veterinary staff.

Gestational age standardized growth parameters for fetal HC, BPD, AC, and FL were evaluated by calculating gestational age specific z-values from normative fetal growth parameters. Linear mixed effects modeling with animal-specific random effects was used to analyze the fetal growth trajectories with advancing gestational age. In order to account for differences in fetal growth parameters at the date of inoculation, changes in fetal growth parameters from date of inoculation (~day 50) were analyzed. That is, changes in fetal growth parameters from date of inoculation were regressed on gestational age (in weeks). An autoregressive correlation structure was used to account for correlations between repeated measurements of growth

parameters over time. The growth trajectories were quantified by calculating the regression slope parameters which were reported along with the corresponding 95% confidence intervals (CI). Fetal growth was evaluated both within and between groups. All reported P-values are two-sided and P<0.05 was used to define statistical significance. Statistical analyses were conducted using SAS software (SAS Institute, Cary NC), version 9.4.

## Viral RNA isolation from blood

Viral RNA was isolated from macaque blood samples as previously described [63,64]. Briefly, plasma was isolated from EDTA-anticoagulated whole blood on the day of collection either using Ficoll density centrifugation for 30 minutes at 1860 x g if the blood was being processed for PBMC, or it was centrifuged in the blood tube at 1400 x g for 15 minutes. The plasma layer was removed and transferred to a sterile 15 ml conical and spun at 670 x g for an additional 8 minutes to remove any remaining cells. Viral RNA was extracted from a 300 μL plasma aliquot using the Viral Total Nucleic Acid Kit (Promega, Madison, WI) on a Maxwell 16 MDx or Maxwell RSC 48 instrument (Promega, Madison, WI).

## Viral RNA isolation from tissues

Tissue samples, cut to 0.5 cm thickness on at least one side, were stored in RNAlater at 4°C for 2–7 days. RNA was recovered from tissue samples using a modification of the method described by Hansen et al., 2013 [65]. Briefly, up to 200 mg of tissue was disrupted in TRIzol (Lifetechnologies) with 2 x 5 mm stainless steel beads using the TissueLyser (Qiagen) for 3 minutes at 25 r/s twice. Following homogenization, samples in TRIzol were separated using Bromo-chloro-propane (Sigma). The aqueous phase was collected, and glycogen was added as a carrier. The samples were washed in isopropanol and ethanol precipitated. RNA was fully resuspended in 5 mM tris pH 8.0.

## Quantitative reverse transcription PCR (QRT-PCR)

vRNA isolated from both fluid and tissue samples was quantified by QRT-PCR as previously described [66]. The RT-PCR was performed using either the SuperScript III Platinum One-Step Quantitative RT-PCR system (Invitrogen, Carlsbad, CA) or Taqman Fast Virus 1-step master mix (Applied Biosystems, Foster City, CA) on a LightCycler 96 or LightCycler 480 instrument (Roche Diagnostics, Indianapolis, IN). Viral RNA concentration was determined by interpolation onto an internal standard curve composed of seven 10-fold serial dilutions of a synthetic ZIKV RNA fragment based on a ZIKV strain derived from French Polynesia that shares >99% similarity at the nucleotide level to the Puerto Rican strain used in the infections described in this manuscript.

## Statistical analysis of viral loads

Plasma viral load curves were generated using GraphPad Prism software. The area under the curve of 0–10 d.p.i. was calculated using GraphPad software and a two-sample t-test was performed to assess differences in the peak, duration, and area under the curve of ZIKV viremia between DENV-immune and DENV-naïve macaques. Duration was calculated both as a raw number of days and as a binary, with >21 days of viremia considered "prolonged" and <21 days considered "non-prolonged." To compare differences in the viral burden in the maternal-fetal interface, a non-parametric Mann-Whitney U test was used to assess differences in each of the maternal-fetal interface tissues. GraphPad Prism 8 software was used for these analyses.

### Plaque reduction neutralization test (PRNT)

Macaque serum was isolated from whole blood on the same day it was collected using a serum separator tube (SST) or clot activator (CA) tube. The SST or CA tube was centrifuged for at least 20 minutes at 1400 x g, the serum layer was removed and placed in a 15 ml conical and centrifuged for 8 minutes at 670 x g to remove any additional cells. Serum was screened for ZIKV neutralizing antibody utilizing a plaque reduction neutralization test (PRNT) on Vero cells as described in [67] against DENV-2 and ZIKV-PR. Neutralization curves were generated using GraphPad Prism 8 software. The resulting data were analyzed by non-linear regression to estimate the dilution of serum required to inhibit 50% and 90% of infection

### Inhibition ELISA (iELISA assay)

The DENV iELISA was performed on serum samples as previously described [2,68,69]. Briefly, ELISA plates were coated with anti-DENV polyclonal IgG to capture a mixture of DENV 1–4 antigen (DENV prototype strains, GenBank Accession #s: KM204119, KM204118, KU050695, KR011349) diluted in Phosphate Buffer Saline + 0.05% Tween 20 at pH 7.4 (PBS-T)[70]. After blocking and additional washes, macaque serum was added in 10-fold serial dilutions (1:10, 1:100, 1:1000, 1:10,000) and incubated for two hours at 37˚C. Thereafter, a set concentration of horseradish peroxidase (HRP)-conjugated polyclonal anti-DENV IgG to each well and incubated for 30 minutes at 37˚C. Following washes, peroxidase substrate TMB was added to wells and incubated for 30 minutes at room temperature, then stopped with sulfuric acid. Plates were read on an ELISA reader, and iELISA titers were estimated relative to negative controls (conjugated antibody only) using the Reed-Muench method [71]. The ZIKV iELISA is similar in design to the DENV iELISA and was performed as described previously [72]. ZIKV-specific monoclonal antibody ZKA64 [73] is used to capture ZIKV antigen prepared as described by [70], macaque serum was added in serial dilutions and competed with HRP-conjugated mAb ZKA64, and iELISA titers were also estimated using the Reed-Muench method.

### ZIKV IgM ELISA

A commercial ZIKV IgM ELISA kit (euroimmun, Lübeck, Germany, EI 2668–9601 M) was used to determine if IgM antibodies were present in fetal serum samples at the time of delivery. The protocol was followed as specified by the manufacturer and all samples were frozen undiluted at -80˚C until use. Duplicates were run for each of the samples. In addition to the positive and negative controls included with the kit, a 14 d.p.i. serum sample from 030–104 was used as a positive control and a 4 d.p.i. serum sample from 030–104 was used as a negative control. All serum samples were diluted 1:100. Immediately upon addition of the stop solution the plate was read at 450 nm. An extinction value was calculated for each sample by calculating the ratio of the OD of the sample to the OD of the assay calibration sample. Per the kit's recommendation, any sample with a ratio >1.1 was considered to be positive for ZIKV IgM, between 1.1 and 0.8 was considered to be borderline, and <0.8 was considered to be negative.

### Cesarean section and tissue collection

Between 159–161 days gestation, infants were delivered via cesarean section and tissues were collected. The fetus, placenta, fetal membranes, umbilical cord, and amniotic fluid were collected at surgical uterotomy, and maternal tissues were biopsied during laparotomy. These were survival surgeries for the dams and offspring. Amniotic fluid was removed from the amniotic sac, then infant was removed from the amniotic sac, the umbilical cord clamped, and

neonatal resuscitation performed as needed. The placenta and fetal membranes were then collected. Infants were placed with their mothers following the dam's recovery from surgery.

Tissues were dissected as previously described [48] using sterile instruments that were changed between each organ and tissue type to minimize possible cross contamination. Each organ/tissue was evaluated grossly, dissected with sterile instruments in a sterile culture dish, and sampled for histology, viral burden assay, and/or banked for future assays. A comprehensive listing of all specific tissues collected and analyzed is presented in Fig 6A (maternal-fetal interface tissues) and S2 Table (maternal biopsies and fetal fluids). Biopsies of the placental bed (uterine placental attachment site containing deep decidua basalis and myometrium), maternal liver, spleen, and a mesenteric lymph node were collected aseptically during surgery into sterile petri dishes, weighed, and further processed for viral burden and when sufficient sample size was obtained, histology.

In order to more accurately capture the distribution of ZIKV in the placenta, each placental disc was separated, fetal membranes sharply dissected from the margin, weighed, measured, and placed in a sterile dish on ice. A 1-cm-wide cross section was taken from the center of each disc, including the umbilical cord insertion on the primary disc, and placed in 4% paraformaldehyde. Individual cotyledons, or perfusion domains, were dissected using a scalpel and placed into individual petri dishes. From each cotyledon, a thin center cut was taken using a razor blade and placed into a cassette in 4% paraformaldehyde. Once the center cut was collected, the decidua and the chorionic plate were removed from the remaining placenta. From each cotyledon, pieces of decidua, chorionic plate, and chorionic villi were collected into two different tubes—one with RNAlater for vRNA isolation and one with 20%FBS/PBS for other virological assays.

## Histology

Following collection, tissues were handled as described previously [63]. All tissues were fixed in 4% paraformaldehyde for 24 hours and transferred into 70% ethanol until processed and embedded in paraffin. Paraffin sections (5 µm) were stained with hematoxylin and eosin (H&E). Pathologists were blinded to vRNA findings when tissue sections were evaluated microscopically. Photomicrographs were obtained using a bright light microscope Olympus BX43 and Olympus BX46 (Olympus Inc., Center Valley, PA) with attached Olympus DP72 digital camera (Olympus Inc.) and Spot Flex 152 64 Mp camera (Spot Imaging) and captured using commercially available image-analysis software (cellSens DimensionR, Olympus Inc. and spot software 5.2).

## Placental histology soring

Pathological evaluation of the cross-sections of each of the individual placental cotyledons were performed by Dr. Terry Morgan who was blinded to experimental condition. Each of the cross sections were evaluated for the presence of chronic histiocytic intervillositis (CHIV), infarctions, villous stromal calcifications, and vasculopathy. A three-way ANOVA was performed to assess statistical significance among groups for each parameter, including placental weight.

Two of three boarded veterinary pathologists, blinded to vRNA findings, independently reviewed the central cross section of each placental disc and quantitatively scored the placentas on 22 independent criteria. Six of the criteria are general criteria assessing placental function, two assess villitis, three criteria assess the presence of fetal vascular malperfusion, and 11 criteria assess the presence of maternal vascular malperfusion. The scoring system was developed by Dr. Michael Fritsch, Dr. Heather Simmons, and Dr. Andres Mejia. A summary table of the

criteria scored and the scale used for each criterion can be found in S2 Table. Once initial scores were assigned, all pathologists met to discuss and resolve any significant discrepancies in scoring. Scores were assigned to each placental disc unless the criteria scored corresponded to the function of the entire placenta.

For criteria measured on a quantitative scale, median scores and interquartile range were calculated for each experimental group. For criteria measured on a binary "present/not present" scale, the cumulative incidence in each experimental group was calculated as a frequency and a percentage. For quantitative criteria, a non-parametric Wilcoxon rank test was used to calculate statistical significance between each of the groups and between the mock-infected group and the two ZIKV-infected groups. For binary features, Fisher's exact test was used to calculate statistical significance between each of the groups and between the mock-infected group and the two ZIKV-infected groups. To determine whether chronic villitis correlated with the criteria assessing fetal malperfusion and whether chronic deciduitis correlated with the criteria assessing maternal malperfusion, scores were adjusted to be on the same scale (i.e., converting measures on a 0–1 scale to a 0–2 scale) so that each parameter carried equal weight in the combined score. A nonparametric Spearman's correlation was used to determine the correlation.

## Supporting information

**S1 Fig. PRNT neutralization curves.** PRNT titers against DENV (**A-C**) and ZIKV (**D-E**) at 28 days post-DENV challenge, 0 days post-ZIKV challenge, and 28–35 days post-ZIKV challenge. (PDF)

**S2 Fig. ZIVK IgM ELISA titer of fetal samples.** An IgM ELISA using ZIKV NS1 antigen was performed on fetal samples taken on the day of delivery. Maternal plasma and serum samples from 042–108 were included as positive (14 d.p.i.) and negative (4 d.p.i.) controls. Internal assay positive and negative controls fell within the expected range defined by the manufacturer. Fetal serum was used when available. In the absence of available fetal serum, fetal plasma was used for 042–502 and umbilical cord plasma was used for 042–501. (PDF)

**S3 Fig. Placental pathology scoring.** The central cross-section of each placental disc was evaluated for 22 pathologic changes. A description of the scoring system can be found in S2 Table. Features specific to the fetal membranes or uterus are noted in disc 1 scoring. Statistical pairwise comparisons between each group were performed for each feature. For quantitative features (1–9, 12–16, 22) a non-parametric Wilcoxon rank sum test was used; for binary features (10–11, 17–21) Fisher's exact test was used. For quantitative features, the median value is shown with error bars representing the interquartile range. When compared to the mock-infected cohort, the DENV-immune macaques had significantly higher scores for transmural infarction (disc 1: p = 0.0371; disc 2: not significant), chronic villitis (disc 1: p = 0.0207; disc 2: p = 0.0151), avascular villi (disc 1: p = 0.0152; disc 2: not significant), and chronic retroplacental hemorrhage (disc 1: p = 0.0152; disc 2: not significant). (PDF)

**S1 Table. Maternal and Fetal Tissue and Fluid ZIKV RNA Detection.** (PDF)

**S2 Table. Placental Pathology Scoring System (central section).** (PDF)

## Acknowledgments

We thank the Veterinary Services, Colony Management, Scientific Protocol Implementation, and the Pathology Services staff at the Wisconsin National Primate Research Center (WNPRC) for their contributions to this study. We thank Brandy Russell for providing virus isolates. The following reagent was obtained through BEI Resources, NIAID, NIH: Dengue Virus Type 2, DENV-2/US/BID-V594/2006, NR-43280.

## Author Contributions

**Conceptualization:** Chelsea M. Crooks, Christina M. Newman, Dawn M. Dudley, David H. O'Connor, Emma L. Mohr, Thaddeus G. Golos, Thomas C. Friedrich, Matthew T. Aliota.

**Data curation:** Chelsea M. Crooks, Andrea M. Weiler, Sierra L. Rybarczyk, Mason I. Bliss, Megan E. Murphy, Heather A. Simmons, Ann M. Mitzey, Elaina Razo, Keisuke Yamamoto, Phoenix M. Shepherd, Terry K. Morgan, Angel Balmaseda, David H. O'Connor, Emma L. Mohr, Thaddeus G. Golos, Thomas C. Friedrich, Matthew T. Aliota.

**Formal analysis:** Chelsea M. Crooks, Heather A. Simmons, Jens C. Eickhoff, Kathleen M. Antony, Terry K. Morgan.

**Funding acquisition:** David H. O'Connor, Emma L. Mohr, Thaddeus G. Golos, Thomas C. Friedrich, Matthew T. Aliota.

**Investigation:** Chelsea M. Crooks, Andrea M. Weiler, Sierra L. Rybarczyk, Mason I. Bliss, Anna S. Jaeger, Megan E. Murphy, Heather A. Simmons, Andres Mejia, Michael K. Fritsch, Jennifer M. Hayes, Ann M. Mitzey, Elaina Razo, Katarina M. Braun, Keisuke Yamamoto, Phoenix M. Shepherd, Amber Possell, Kara Weaver, Kathleen M. Antony, Terry K. Morgan, Nancy Schultz-Darken, Eric Peterson, Angel Balmaseda, Matthew T. Aliota.

**Methodology:** Chelsea M. Crooks, Heather A. Simmons, Andres Mejia, Michael K. Fritsch, Jens C. Eickhoff, Terry K. Morgan, Christina M. Newman, Dawn M. Dudley, Leah C. Katzelnick, Angel Balmaseda, Eva Harris, David H. O'Connor, Emma L. Mohr, Thaddeus G. Golos, Thomas C. Friedrich, Matthew T. Aliota.

**Project administration:** Chelsea M. Crooks, Megan E. Murphy, Jennifer M. Hayes, Ann M. Mitzey, Elaina Razo, Elizabeth A. Brown, Keisuke Yamamoto, Phoenix M. Shepherd, Dawn M. Dudley, Nancy Schultz-Darken, Eric Peterson, Leah C. Katzelnick, Eva Harris, David H. O'Connor, Emma L. Mohr, Thaddeus G. Golos, Thomas C. Friedrich, Matthew T. Aliota.

**Resources:** David H. O'Connor, Emma L. Mohr, Thaddeus G. Golos, Thomas C. Friedrich, Matthew T. Aliota.

**Software:** Jens C. Eickhoff, David H. O'Connor.

**Supervision:** David H. O'Connor, Emma L. Mohr, Thaddeus G. Golos, Thomas C. Friedrich, Matthew T. Aliota.

**Validation:** Chelsea M. Crooks, Andrea M. Weiler, Sierra L. Rybarczyk, Mason I. Bliss, Megan E. Murphy, Heather A. Simmons, Andres Mejia, Michael K. Fritsch.

**Visualization:** Chelsea M. Crooks, Jens C. Eickhoff.

**Writing – original draft:** Chelsea M. Crooks.

**Writing – review & editing:** Chelsea M. Crooks, Andrea M. Weiler, Sierra L. Rybarczyk, Mason I. Bliss, Anna S. Jaeger, Megan E. Murphy, Heather A. Simmons, Andres Mejia, Michael K. Fritsch, Jennifer M. Hayes, Jens C. Eickhoff, Ann M. Mitzey, Elaina Razo,

Katarina M. Braun, Elizabeth A. Brown, Keisuke Yamamoto, Phoenix M. Shepherd, Amber Possell, Kara Weaver, Kathleen M. Antony, Terry K. Morgan, Christina M. Newman, Dawn M. Dudley, Nancy Schultz-Darken, Eric Peterson, Leah C. Katzelnick, Angel Balmaseda, Eva Harris, David H. O'Connor, Emma L. Mohr, Thaddeus G. Golos, Thomas C. Friedrich, Matthew T. Aliota.

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
