## [Decision Letter · Decision Letter 0]

6 Apr 2021

Dear Dr. Aliota,

Thank you very much for submitting your manuscript "Prior dengue immunity enhances Zika virus infection of the maternal-fetal interface in rhesus macaques" for consideration at PLOS Neglected Tropical Diseases. As with all papers reviewed by the journal, your manuscript was reviewed by members of the editorial board and by several independent reviewers. In light of the reviews (below this email), we would like to invite the resubmission of a significantly-revised version that takes into account the reviewers' comments. 

We cannot make any decision about publication until we have seen the revised manuscript and your response to the reviewers' comments. Your revised manuscript is also likely to be sent to reviewers for further evaluation.

Sincerely,

Paulo F. P. Pimenta, Ph.D

Deputy Editor

Paulo Pimenta

Deputy Editor

Reviewer's Responses to Questions

**Key Review Criteria Required for Acceptance?**

**Methods**

-Are the objectives of the study clearly articulated with a clear testable hypothesis stated?

-Is the study design appropriate to address the stated objectives?

-Is the population clearly described and appropriate for the hypothesis being tested?

-Is the sample size sufficient to ensure adequate power to address the hypothesis being tested?

-Were correct statistical analysis used to support conclusions?

-Are there concerns about ethical or regulatory requirements being met?

Reviewer #1: Although it is an interesting, unprecedented study, it has some important limitations. A small number of animals was enrolled. Only one DENV serotype (DENV-2) was tested. In addition, differences between the periods from DENV inoculation to ZIKV challenge in each individual were observed (up to 35 days).

Reviewer #2: Yes, to all of the above questions.

Reviewer #3: Both assays and assay description should be carefully examined here - terms such as "exposure" and "infection" can be interpreted loosely and infection measurements by RNA quantification must be backed up by other experimental results. Additional assays to explain the discrepancy between increased "infection" or viremia at the maternal-fetal interface but no increased "infection" by viral RNA in plasma need to be done. 

Both DENV and ZIKV antibodies should be quantified from the offspring and mothers at various timepoints post-birth.

Reviewer #4: The study objective has been clearly stated. Authors sought to investigate the impact of pre-existing dengue virus immunity on Zika virus infection and pathogenesis during pregnancy by using the nonhuman primate model. The number of animals are small per each group. However, this study utilizes rhesus macaques and involves infections during pregnancy. That can be a constraint on increasing the animals numbers in the study. Regulatory requirements are met for this study.

**Results**

-Does the analysis presented match the analysis plan?

-Are the results clearly and completely presented?

-Are the figures (Tables, Images) of sufficient quality for clarity?

Reviewer #1: The authors observed a significantly greater burden of ZIKV RNA in the chorionic plate in DENV-immune macaques as compared to DENV-naïve macaques as well as an association between prolonged viremia and the presence of ZIKV vRNA in the maternal-fetal interface. Since previous evidence suggest that the elapsed time from the DENV exposure to the ZIKV infection may influence in the risk of congenital Zika syndrome (Carvalho MS et al. Association of past dengue fever epidemics with the risk of Zika microcephaly at the population level in Brazil. Sci Rep. 2020;10:1752), the comparison between groups of macaques with different elapsed times between DENV and ZIKA infections would be useful.

Reviewer #2: Yes, to all of the above questions.

Reviewer #3: Lines 181-192 

This paragraph is extremely confusing, to the point of not having a decipherable meaning. First, the word "fell" on Line 181 makes no sense here. The statement that DENV ELISA titers were up 4-fold in 6 of 8 and 4 of 4 animals is not in any context. The statements "cross-reactive ZIKV titers were stable or increased" then followed by "cross-reactive ZIKV antibodies were undetectable in 3 of 4 macaques" are not compatible. This entire section needs to be rewritten in a clear, concise manner so that results can be interpreted and conclusions can be drawn.

Reviewer #4: The results are presented well. The figures have provided data points for all the animals in the study. Immunological assays for several time points were not accomplished due to COVID-19 limitations.

**Conclusions**

-Are the conclusions supported by the data presented?

-Are the limitations of analysis clearly described?

-Do the authors discuss how these data can be helpful to advance our understanding of the topic under study?

-Is public health relevance addressed?

Reviewer #1: The work by Crooks et al. entitled “Prior dengue immunity enhances Zika virus infection of the maternal-fetal interface in rhesus macaques” evaluated the impact of prior DENV immunity over the infection of maternal-fetal interface tissues with ZIKV in macaques. This is the first study to assess the aforementioned relationship in a translational non-human primate model.

Reviewer #2: Yes, to all of the above questions.

Reviewer #3: The inability to examine fetal tissues results in the inability to claim "no vertical transmission". There is also no mechanism proposed or examined to explain the disconnect between the increased infection in maternal-fetal interfaces and lack of enhanced viremia in maternal plasma. In addition, other assays are necessary (immune profiling - T cell, innate response, cytokine profiling; disease severity indicators, etc) in order to claim whether there are signs of enhancement or not.

Reviewer #4: Authors need to add sufficient justification about the pathogenic potential of Dengue and Zika virus strains used in this study for infections in rhesus macaques. Animal numbers in experimental groups are small. Authors have stated some of the limitations of the study.

**Editorial and Data Presentation Modifications?**

Reviewer #1: I would like to suggest some changes in the manuscript that may aid in the improvement of the article quality. I suggest that the authors add an explanation on why the 45th gestational day was chosen for ZIKV challenges (line 136). In addition, I suggest that the paragraph from the line 154 to the line 161 be summarized or even excluded and that the information contained in that part of the text be placed in the introduction section and/or in the discussion section of the manuscript.

Reviewer #2: (No Response)

Reviewer #3: (No Response)

Reviewer #4: No data are presented on the levels of inflammatory cytokines or monocyte activation during viral infections.

**Summary and General Comments**

Reviewer #1: The work by Crooks et al. entitled “Prior dengue immunity enhances Zika virus infection of the maternal-fetal interface in rhesus macaques” evaluated the impact of prior DENV immunity over the infection of maternal-fetal interface tissues with ZIKV in macaques. This is the first study to assess the aforementioned relationship in a translational non-human primate model. The authors observed a significantly greater burden of ZIKV RNA in the chorionic plate in DENV-immune macaques as compared to DENV-naïve macaques as well as an association between prolonged viremia and the presence of ZIKV vRNA in the maternal-fetal interface. Although it is an interesting, unprecedented study, it has some important limitations. A small number of animals was enrolled. Only one DENV serotype (DENV-2) was tested. In addition, differences between the periods from DENV inoculation to ZIKV challenge in each individual were observed (up to 35 days). Moreover, since previous evidence suggest that the elapsed time from the DENV exposure to the ZIKV infection may influence in the risk of congenital Zika syndrome (Carvalho MS et al. Association of past dengue

727 fever epidemics with the risk of Zika microcephaly at the population level in Brazil. Sci Rep. 2020;10:1752), the comparison between groups of macaques with different elapsed times between DENV and ZIKA infections would be useful.

Reviewer #2: The authors conducted a study to assess if the pre-existing immunity to dengue virus could enhance Zika virus disease by the antibody-dependent enhancement phenomenon.

The proposed theme is of great relevance to understanding the risks of congenital malformation in children whose mothers have been infected with Zika virus, especially in areas endemic to other flaviviruses. This hypothesis is based on studies that suggest this phenomenon's occurrence in vitro and animal models, but which has been widely contested as a hypothesis to justify the worsening of secondary cases of Dengue virus infection in humans.

The writing of the article is excellent and objective. The experiments were carried out with sophisticated methods and techniques and with high technical rigor. The data generated has been correctly analyzed and appears to be robust.

The results suggest that the pre-existing DENV immunity does not cause fetal abnormalities or even alter the Zika virus replication in maternal plasma.

Therefore, I am in favor of publishing the article in its present form.

Reviewer #3: This study focuses on the important and relevant topic of the impact of pre-existing dengue virus immunity on Zika virus infection in pregnant individuals using non-human primate models of infection. While the questions asked are significant and of high interest, the studies are incomplete and experiments too preliminary to justify conclusions.

Reviewer #4: In this manuscript, Authors report that prior dengue immunity enhances Zika virus infection of the maternal-fetal interface in rhesus macaques. This study has examined the impact of pre-existing dengue virus immunity on Zika virus infection and pathogenesis during pregnancy in the rhesus macaque model. Authors found that prior DENV-2 exposure enhanced ZIKV infection of maternal-fetal interface tissues in macaques. However, pre-existing DENV immunity had no detectable impact on ZIKV replication kinetics in maternal plasma, and all pregnancies progressed to term without adverse outcomes or gross fetal abnormalities detectable at delivery. 

Coinfections with dengue and Zika viruses have been reported in endemic regions. Therefore, it is important to understand the impact of re-existing immunity to any of these viruses on the outcomes of co-infections. It is of special interest to determine the impact of co-infections during pregnancy and on fetal growth and health. The manuscript reports that preexisting immunity to DENV leads to increased Zika virus infection in placenta. However, this did not lead to increased fetal abnormalities. Findings in this manuscript are very informative.

Data in Figure 2 demonstrate that pre-existing DENV-2 immunity did not increase or alter ZIKV replication kinetics during gestation. These data are consistent with previous reports of DENV and ZIKV co-infections in humans and non-human primates which did not result in increased viral replication or enhancement of disease pathogenesis. Findings in this manuscript show that pre-existing DENV immunity did not dampen or protect against ZIKV infection. 

Antibodies cross-reactive with ZIKV were found in DENV-immune animals. However, that did not interfere with the magnitude of antibody response to ZIKV. These findings are important since they showed that pre-existing cross-reactive DENV antibodies did not impair host immune response to ZIKV. Samples were not analyzed for several time points (due to COVID19 issues). 

There was no evidence of vertical transmission of ZIKV in infants. QRT-PCR data are presented to show low levels of ZIKV RNA in placental samples. Although the assay may detect low levels of viral RNA in some of the animals from both groups (DENV-immune and DENV-naïve), data are not convincing to sufficiently demonstrate that there was enhanced infection at maternal-fetal interface. No infectious virus was recovered. Additionally, viral RNA levels are too low and the animal numbers per each group are not sufficient. 

These viral infections are known to induce inflammatory cytokine expression and monocyte activation. Considering the minimal impact of viral infections on the animals in this study, assessment of these parameters could determine whether the viral infections had any substantial effect on the host. 

There was no demonstrable placental pathology due to ZIKV infection in the present study. This raises the question about the pathogenic potential of the viral strains used to infect animals in the study. Authors need to comment on this issue.

PLOS authors have the option to publish the peer review history of their article (what does this mean?). If published, this will include your full peer review and any attached files.

Reviewer #1: No

Reviewer #2: Yes: Breno de Mello Silva

Reviewer #3: No

Reviewer #4: No
---

## [Decision Letter · Decision Letter 1]

9 Jul 2021

Dear Dr. Aliota,

We are pleased to inform you that your manuscript 'Previous exposure to dengue virus is associated with increased Zika virus burden at the maternal-fetal interface in rhesus macaque' has been provisionally accepted for publication in PLOS Neglected Tropical Diseases.

Best regards,

Paulo F. P. Pimenta, Ph.D

Deputy Editor

Paulo Pimenta

Deputy Editor

Reviewer's Responses to Questions

**Key Review Criteria Required for Acceptance?**

**Methods**

-Are the objectives of the study clearly articulated with a clear testable hypothesis stated?

-Is the study design appropriate to address the stated objectives?

-Is the population clearly described and appropriate for the hypothesis being tested?

-Is the sample size sufficient to ensure adequate power to address the hypothesis being tested?

-Were correct statistical analysis used to support conclusions?

-Are there concerns about ethical or regulatory requirements being met?

Reviewer #1: The answers to the reviewers were pertinent and adjustments were made.

Reviewer #2: Yes, to all of the above questions.

Reviewer #3: (No Response)

Reviewer #4: The objectives of the study are identified. The major limitation of this study is the small number of animals per each experimental group. The concern about the sample size was raised in the previous review which has not been resolved in the revised manuscript. However, Authors have clearly identified and described limitations of this study in the revised text. Since the study involves non-human primates, it is challenging to address this limitation for the current manuscript.

The limitation of small animal numbers could have been partly overcome by longitudinal assessments of immune and cytokine responses against the virus in mothers and their offspring. However, Authors have responded that those data will be included in a future manuscript and are not part of the current manuscript.

There are no concerns regarding ethical or regulatory requirements.

**Results**

-Does the analysis presented match the analysis plan?

-Are the results clearly and completely presented?

-Are the figures (Tables, Images) of sufficient quality for clarity?

Reviewer #1: The answers to the reviewers were pertinent and adjustments were made.

Reviewer #2: Yes, to all of the above questions.

Reviewer #3: (No Response)

Reviewer #4: The Results section has been revised in response to the Reviewers’ comments and most concerns have been addressed. Results are well presented in the Figures with clarity.

**Conclusions**

-Are the conclusions supported by the data presented?

-Are the limitations of analysis clearly described?

-Do the authors discuss how these data can be helpful to advance our understanding of the topic under study?

-Is public health relevance addressed?

Reviewer #1: The answers to the reviewers were pertinent and adjustments were made.

Reviewer #2: Yes, to all of the above questions.

Reviewer #3: (No Response)

Reviewer #4: The conclusions are in general, supported by the data. Authors have presented data to show an increased level of ZIKV RNA levels at the maternal fetal interface tissue in DENV immune macaques. The text has been revised to strengthen their conclusion.

A better description has been provided in the revised text about the infectivity and pathogenicity of ZIKV strain used in the present study. Authors have stated that the ZIKV strain used in this study seems to be less pathogenic than the ZIKV strains used by other investigators.

Authors have described the relevance of the translational nonhuman model in the study of DENV an ZIKV infections and their impact at the maternal fetal interface.

**Editorial and Data Presentation Modifications?**

Reviewer #1: The answers to the reviewers were pertinent and adjustments were made. I suggest manuscript approval.

Reviewer #2: (No Response)

Reviewer #3: (No Response)

Reviewer #4: Accept

**Summary and General Comments**

Reviewer #1: The answers to the reviewers were pertinent and adjustments were made.

Reviewer #2: I continue to consider the article very well planned, with exciting results, robust and pertinent to this study area. The authors responded convincingly to all the points raised by the other reviewers.

Reviewer #3: The authors have addressed all of the reviewers' comments thoroughly and thoughtfully - and have added explanations and data to the manuscript to imrpove clarity and aid in interpretation. I have no hesitation in recommending acceptance at this point.

Reviewer #4: In this revised manuscript, Authors report that previous exposure to dengue virus is associated with increased Zika virus burden at the maternal-fetal interface in rhesus macaques. The manuscript has been revised to adequately address most of the previous Reviewers’ comments.

Authors have utilized a preclinical nonhuman primate model to examine the impact of pre-existing DENV immunity on ZIKV infection during pregnancy and on pregnancy outcomes. The study reports lack of adverse effects of ZIKV infection on pregnancies and absence of fetal abnormalities in DENV immune rhesus macaques.

Several issues about this manuscript are not resolved. The animal numbers per each group are small. Immune and cytokine analyses are not completed. Zika virus strain used in the study seems less pathogenic compared to Zika virus stains used in other nonhuman primate studies. This adds further challenge to the outcome analysis in the groups with small number of animals. Despite all these limitations, the manuscript provides useful information on the nonhuman primate model for investigating the impact of preexisting immunity to DENV on ZIKV infection as well as DENV and ZIKV coinfections. The revised title is appropriate and is reflective of the findings in the manuscript. The manuscript is also strengthened by the clear description about the limitations of the present study.

PLOS authors have the option to publish the peer review history of their article (what does this mean?). If published, this will include your full peer review and any attached files.

Reviewer #1: No

Reviewer #2: No

Reviewer #3: No

Reviewer #4: No

---

## [Editor Report · Acceptance letter]

26 Jul 2021

Dear Dr. Aliota,

We are delighted to inform you that your manuscript, "Previous exposure to dengue virus is associated with increased Zika virus burden at the maternal-fetal interface in rhesus macaques," has been formally accepted for publication in PLOS Neglected Tropical Diseases.

Best regards,

Shaden Kamhawi

co-Editor-in-Chief

Paul Brindley

co-Editor-in-Chief
